# Deletion of neural estrogen receptor alpha induces sex differential effects on reproductive behavior in mice

Anne-Charlotte Trouillet [1], Suzanne Ducroq[1], Lydie Naulé[1], Daphné Capela[1], Caroline Parmentier[1], Sally Radovick[2], Hélène Hardin-Pouzet[1] & Sakina Mhaouty-Kodja [1✉]

Estrogen receptor (ER) α is involved in several estrogen-modulated neural and peripheral functions. To determine its role in the expression of female and male reproductive behavior, a mouse line lacking the *ERα* in the nervous system was generated. Mutant females did not exhibit sexual behavior despite normal olfactory preference, and had a reduced number of progesterone receptor-immunoreactive neurons in the ventromedial hypothalamus. Mutant males displayed a moderately impaired sexual behavior and unaffected fertility, despite evidences of altered organization of sexually dimorphic populations in the preoptic area. In comparison, males deleted for both neural *ERα* and androgen receptor (*AR*) displayed greater sexual deficiencies. Thus, these data highlight a predominant role for neural *ERα* in females and a complementary role with the AR in males in the regulation of sexual behavior, and provide a solid background for future analyses of neuronal versus glial implication of these signaling pathways in both sexes.

[1] Sorbonne Université, CNRS, INSERM, Neuroscience Paris Seine – Institut de Biologie Paris Seine, 75005 Paris, France. [2] Unit of Pediatric Endocrinology, Department of Pediatrics, Rutgers-Robert Wood Johnson Medical School, New Brunswick, NJ, USA. ✉email: sakina.mhaouty-kodja@sorbonne-universite.fr

Sex steroids play a key role in the developmental organization and adult activation of the neural circuitry underlying reproductive behavior. In males, it is largely accepted that perinatal testosterone and its neural metabolite estradiol irreversibly masculinize and defeminize the neural structures underlying behavioral responses[1]. During this period, the ovaries are inactive and the α-fetoprotein protects the female brain from these masculinizing effects by selectively binding to maternal and male sibling-derived estradiol[2]. The ovaries start secreting estradiol around postnatal day 7[3]. It has been suggested that estradiol, during the time window between postnatal days 15 and 25, is important for a full expression of female sexual behavior during adulthood[4,5]

In the nervous system, testosterone activates the androgen receptor (AR), while estradiol acts mainly through two nuclear estrogen receptors (ER) α and β encoded by two different genes, with increasing evidence indicating the potential involvement of other non-genomic pathways as well[6]. Conditional genetic mouse models were generated to determine the neural role of nuclear AR and ERs in the expression of reproductive behavior, without interfering with their peripheral functions. We have previously shown that early AR deletion in neuronal and glial cells impaired the expression of sexual and aggressive behaviors in males[7,8]. In particular, neural AR deletion did not affect the perinatal organization of known sexually dimorphic populations in the preoptic area, the key hypothalamic region controlling male sexual behavior[9,10], but was found to alter the postnatal organization of the spinal nuclei involved in erection and ejaculation[11,12]. Further neural deletion of ERβ, using the same transgene, showed that this receptor is not involved in the expression of male or female sexual behavior, but rather mediates estrogen-induced regulation of social and aggressive behaviors in males and mood-related behavior in both sexes[13–15]. In this context, the reproductive phenotype induced by early neural ERα deletion using this same transgene still needs to be determined. Previous studies addressing the effects of ERα deletion in neurons[16,17], or in specific neuronal populations such as glutamatergic and GABAergic neurons[18], or Tac2 or kisspeptin neurons[19–22] only focused on the regulation of the hypothalamic-pituitary-gonadal axis in females. In a male study, it has been shown that postnatal ERα deletion in vesicular GABA transporter (Vgat) neurons altered mating and territorial behaviors without changing hormonal levels[23], while its deletion in vesicular glutamate transporter (Vglut) 2 neurons increased testosterone levels but did not affect these behaviors. Other studies conducted on males targeted life stages after the perinatal period. A study using viral-mediated knockdown of ERα in specific brain areas reported that pubertal ERα activation in the medial amygdala is crucial for male behavior expression, while adult ERα expression in the medial preoptic and ventromedial nuclei facilitates this behavior[24,25].

The present study was therefore conducted to investigate the effects of neural ERα deletion on male and female behaviors. This mouse model was generated using the nestin-Cre transgene, which triggers gene deletion in neural precursor cells by embryonic day 10.5, before gonadal differentiation, as previously reported for neural deletions of AR and ERβ[8,14]. For this purpose, the impact of ERα^NesCre mutation was assessed in males and females on reproductive parameters including hormonal levels, weight of androgen- or estrogen-dependent organs, sperm count, presence of corpora lutea and fertility. We also assessed the behaviors related to reproduction (mating, olfactory preference, emission of ultrasonic vocalizations), locomotor activity, anxiety-like behavior and aggression. The neuroanatomical organization of sexually dimorphic hypothalamic populations related to male or female sexual behaviors were analyzed. Finally, a double neural knockout for AR and ERα (AR::ERα^NesCre) was generated and the behavioral effects of deletion of both receptors were analyzed in males.

## Results

### Selective neural ERα deletion and physiological parameters in males and females

The selective neural deletion of ERα is shown by the presence of the deleted ERα allele of 223 bp in the brain but not in the ovary of floxed mice expressing the Cre recombinase (Fig. 1a and Supplementary Fig. 1). RT-PCR analysis showed comparable ERα mRNA levels in the ovary and testis of control and mutant animals and reduced levels in the cortex, hippocampus, and hypothalamus of mutant females (−71%, −71%, and −78%, respectively) and males (−75%, −87%, and −84%, respectively) in comparison with their control littermates (Fig. 1b, c).

Mutant females were found to have unchanged body weight, reduced relative weight of ovaries (−25%) and a higher relative weight of the uterus (+170%), an estrogen-sensitive tissue, as compared with the control group (Supplementary Table 1). In agreement with the latter result, circulating levels of estradiol were significantly increased in mutant females (+61% versus controls). Histological analysis showed no corpora lutea in the

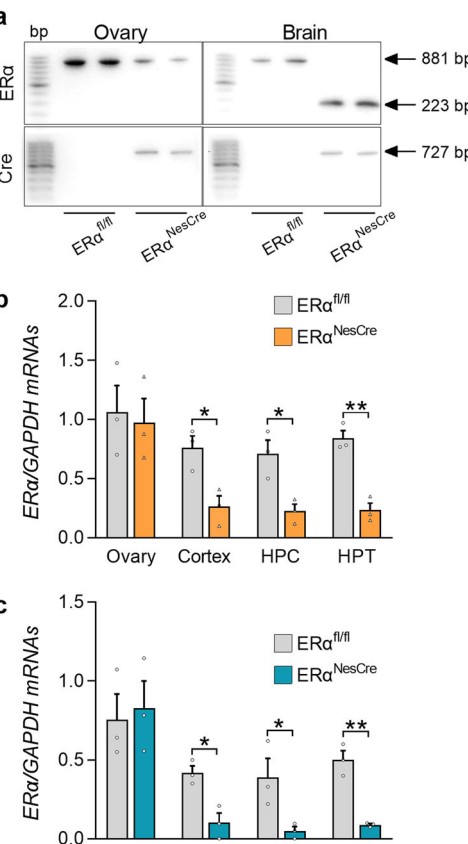

**Fig. 1 Selective neural ERα deletion. a**. Representative PCR analyses performed on ovaries and brains from control (ERα^fl/fl) and mutant females (ERα^NesCre). The upper panels show the presence of the floxed ERα allele (881 bp) in the ovaries of both genotypes and in the brain of controls, and the deleted allele (223 bp) in the brain of mutant females expressing the Cre-recombinase in the lower panels. DNA size markers at 100 bp increments are shown in the left column of the four panels. **b**, **c** Levels of ERα mRNA normalized to Gapdh levels in the ovary, cortex, hippocampus (HPC) and hypothalamus (HPT) of females (**b**), or in the testis, cortex, HPC, and HPT of males (**c**). Data are means ± S.E.M. of three mice per genotype (*$p < 0.05$, **$p < 0.01$ versus controls).

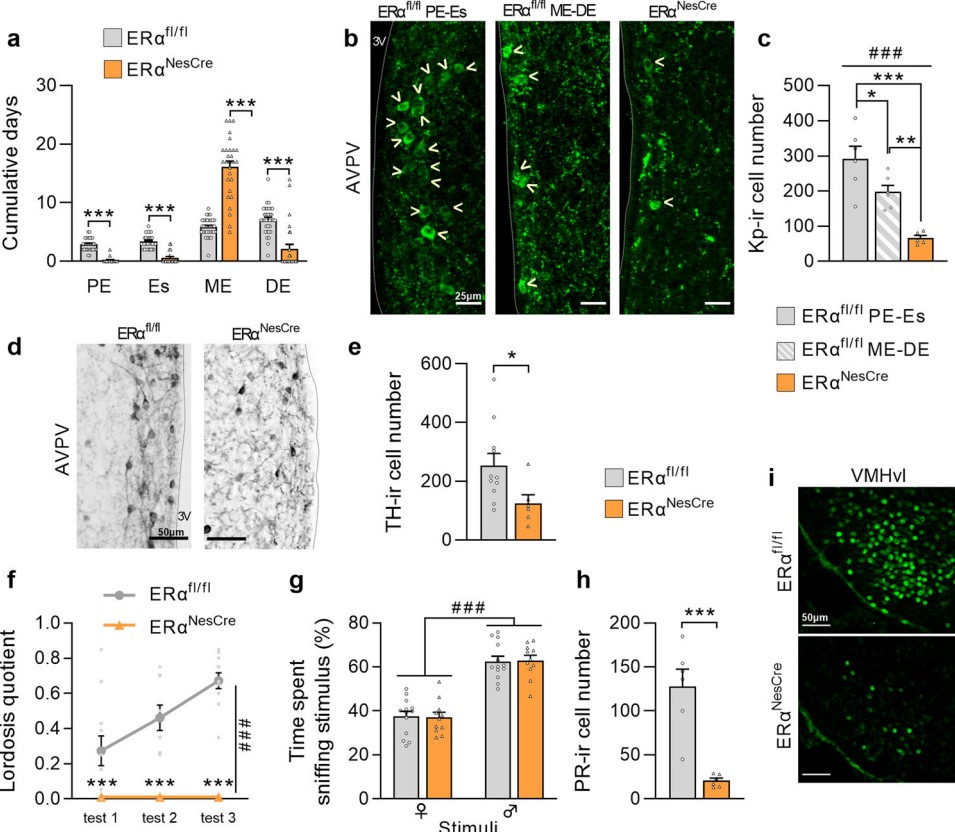

**Fig. 2 Neural _ERα_ deletion induced severe deficit of female reproductive function and behavior. a** Cumulative days in stages of the estrous cycle for control (ERα^fl/fl) and mutant females (ERα^NesCre). PE proestrus, Es estrus, ME metestrus, DE diestrus. Data are means ± S.E.M. for 28–35 females per genotype; ***$p < 0.001$ versus controls. **b, c** Representative images of kisspeptin-immunoreactivity in the anteroventral periventricular area (AVPV) of control females at PE-Es or ME-DE and of mutant littermates (**b**). Kisspeptin positive neurons are highlighted by yellow arrow heads (scale bars = 25 μm), and quantitative data for kisspeptin-ir cell number (**c**) in 5–6 females per group. General effect analyzed by ANOVA (###$p < 0.001$ or #$p < 0.05$) and post hoc analyses (*$p < 0.05$, **$p < 0.01$, ***$p < 0.001$) versus the indicated group are represented. **d, e** Representative images of tyrosine hydroxylase (TH)-immunoreactivity (ir) in the AVPV of control and mutant females (scale bars = 50 μm) (**d**), and quantitative data for TH-ir cell number (**e**) in 6–11 females per genotype. *$p < 0.05$ versus the control group. **f** Lordosis quotient measured in ovariectomized and hormonally primed females ($n = 11$–13 per genotype) over three tests with a week interval between two consecutive tests. Genotype effect (###$p < 0.001$) and posthoc analyses (*$p < 0.05$, ***$p < 0.001$) versus controls are indicated. **g** Olfactory preference of sexually experienced females ($n = 11$–13 per genotype), expressed as percentage of total time spent sniffing male or female stimuli in a Y-maze. The stimulus effect analyzed by two-way ANOVA (###$p < 0.001$) is indicated. **h, i** Quantitative data (**h**) and representative image of progesterone receptor (PR)-ir in the ventromedial hypothalamus (VMH) (**i**), $n = 6$ per genotype, ***$p < 0.001$ versus controls. Scale bars = 50 μm. All data are shown as means ± S.E.M. with individual values.

ovaries of mutant females (Supplementary Table 1). In mutant males, body weight was reduced (−5.5%). The relative weight of the seminal vesicles, which are androgen-sensitive, and testosterone levels were increased (+20% and +100%, respectively), while estradiol levels were reduced (−39%) in comparison with control mice (Supplementary Table 1). The other reproductive parameters, such as testis weight or epididymal sperm count, were unchanged between the two genotypes.

**Arrested estrous cycle and abolished sexual behavior in neural _ERα_ knockout females.** Analysis of the estrous cycle over three weeks showed that mutant females spent more time in the metestrus stage (+174%) and less time (−70 to −93%) in the other stages in comparison with controls (Fig. 2a). The estrous cycle arrest was associated with a reduced number of kisspeptin-immunoreactive neurons in the anteroventral periventricular area (AVPV) of mutant females (−78% versus controls at the proestrus-estrus and −68% versus controls at the metestrus-diestrus) (Fig. 2b, c). For the other sexually dimorphic population of the AVPV, the number of tyrosine hydroxylase (TH)-expressing neurons is shown for control females regardless

of their estrous stage (Fig. 2d, e) since no significant differences were seen between the metestrus-diestrus and proestrus-estrus stages as previously reported[26,27]. Neural _ERα_ deletion reduced by 46% the number of TH-immunoreactive cells in mutant females in comparison with controls (Fig. 2e).

Behavioral analyses were performed on ovariectomized females primed with estradiol and progesterone to induce their receptivity under comparable hormonal conditions. Mutant females never exhibited lordosis posture in response to male mounts over three tests, while their control littermates displayed an increased lordosis quotient over tests reaching 0.7 at the last test (Fig. 2f). This alteration was not due to an effect of neural _ERα_ deletion on olfactory preference since females investigated males more than females regardless of their genotype (Fig. 2g). The impaired lordosis behavior of mutant females was associated with a lower number of progesterone receptor (PR)-immunoreactive neurons (-84% versus controls) in the ventromedial hypothalamus as shown in Fig. 2h.

**Neural _ERα_ deletion reduced but did not abolish male sexual behavior.** All males initiated and achieved sexual behavior in

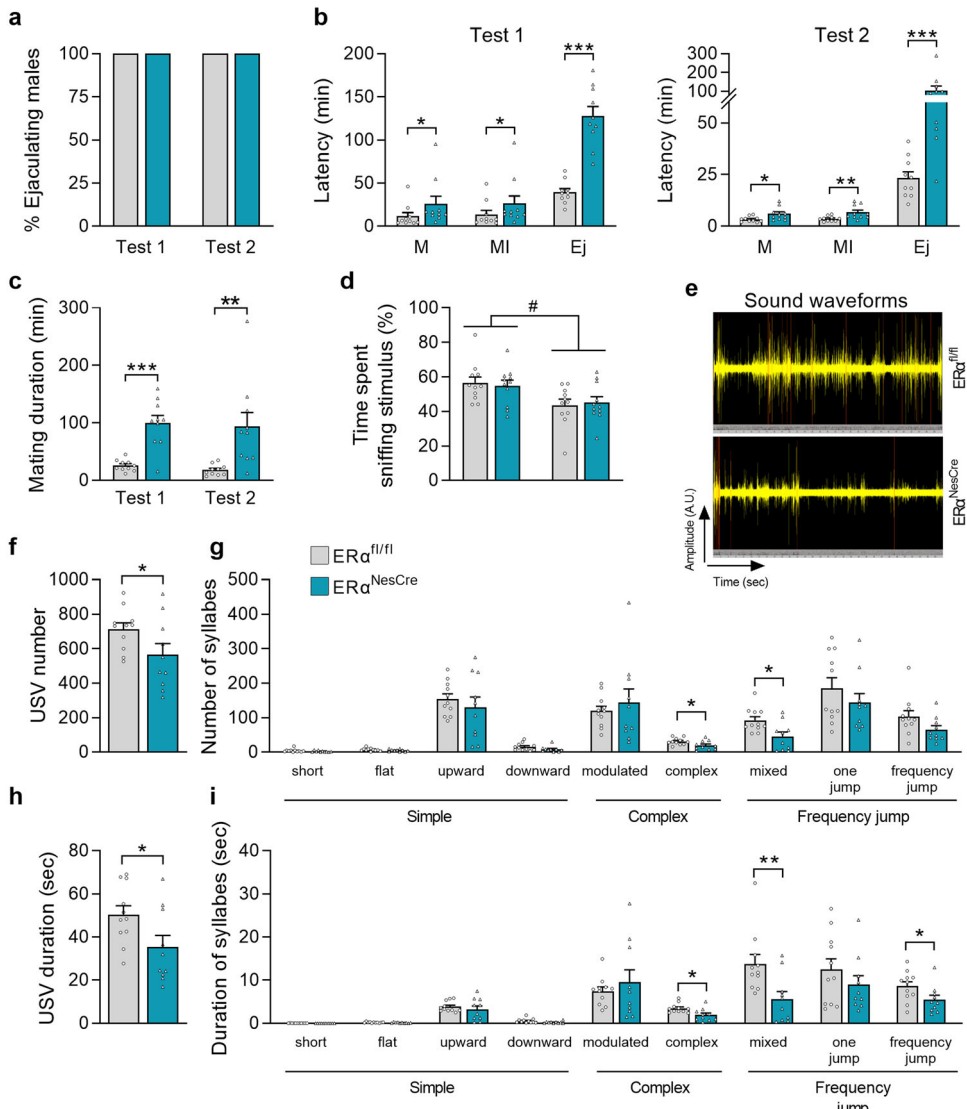

**Fig. 3 The effects of neural ERα^NesCre deletion on sexual behavior in males. a** Percentage of control (ERα^fl/fl) and mutant males (ERα^NesCre) reaching ejaculation in tests 1 (naive animals) and 2 (sexually experienced). Data are shown as percentages. **b** Latencies to mount (M), intromit (MI) and to reach ejaculation (Ej) for control and mutant males ($n = 10$ per genotype) on Test 1 and Test 2. Significant differences between the two genotype are indicated; *$p < 0.05$, **$p < 0.01$, ***$p < 0.001$ versus controls. **c** Mating duration from the first mount to ejaculation was higher in mutant males (***$p < 0.001$ and **$p < 0.01$ in Tests 1 and 2, respectively) ($n = 10$ per genotype). **d** Olfactory preference of sexually experienced control and mutant male mice ($n = 11$ per genotype), expressed as the percentage of total time spent sniffing the stimuli. The stimulus effect (#$p < 0.05$) is indicated. **e** Representative sound waveform for control and mutant males in the presence of a sexually receptive female during a 4-min test. **f, g** Total number of emitted ultrasonic vocalizations USV (**f**) and number of each syllable (**g**) produced by sexually experienced males ($n = 10–11$ per genotype). **h, i** Total duration of emitted USV (**h**) and duration of each syllable (**i**). *$p < 0.05$, or **$p < 0.01$ versus controls ($n = 10–11$ per genotype). Instead stated otherwise, data are shown as means ± S.E.M. with individual values.

both tests 1 and 2 regardless of their genotype (Fig. 3a). However, differences between controls and mutants were seen in the latency to initiate mounting, intromissions, and to reach ejaculation at both tests (Fig. 3b). In accordance, the mating duration was higher in mutant males (+287% and +420% in Tests 1 and 2, respectively) as shown in Fig. 3c. Quantification of the behavioral events showed that mutant males exhibited more mount attempts with or without intromissions and pelvic thrusts (1.8- to 2.4-fold above controls) in both tests (Table 1). These findings together with the increased interval between two consecutive intromissions (+85% and +265% above controls at tests 1 and 2, respectively) suggest a lower behavior of mutant males.

In the olfactory preference test, there was an effect of stimulus but not of genotype, with all males spending more time investigating the female stimulus (Fig. 3d). Analysis of the emission of courtship ultrasonic vocalizations in the presence of a sexually receptive female showed a lower number (−21%) and duration (−30%) of ultrasonic vocalizations in mutant males in comparison with the control group (Fig. 3e, f, h). Detailed quantification of the nine emitted syllables showed a reduced number and duration of the complex, mixed, and frequency-jump syllables (Fig. 3g, i).

At the neuroanatomical level, the effects of neural *ERα* deletion were investigated in two sexually dimorphic neuronal populations known to be organized by perinatal estradiol in the medial preoptic area[28,29]. The number of calbindin-immunoreactive neurons was reduced (−40%; Fig. 4a, b), while that of TH-immunoreactive neurons was higher (+80%; Fig. 4c, d) in mutant

| Table 1 Sexual behavior of gonadally intact ERα$^{NesCre}$ males. | | | | |
|---|---|---|---|---|
| **Behavioral events** | **Sexually naive (Test 1)** | | **Sexually experienced (Test 2)** | |
| | ERα$^{fl/fl}$ | ERα$^{NesCre}$ | ERα$^{fl/fl}$ | ERα$^{NesCre}$ |
| Number of Mounts (M) | 6.4 ± 1.7 | 15.2 ± 3.2* | 5.7 ± 1.8 | 13.3 ± 2.8* |
| Number of Mounts with intromissions (MI) | 26.6 ± 4.7 | 52.5 ± 7.1** | 25.6 ± 4.7 | 45.2 ± 7.1* |
| Intromission ratio MI/M + MI (%) | 78 ± 5 | 76 ± 4 | 82 ± 5 | 78 ± 3 |
| MI duration (s) | 17.9 ± 2.1 | 17.6 ± 1.2 | 18.6 ± 1.7 | 19.6 ± 1.5 |
| MI interval (s) | 43.7 ± 7.9 | 80.9 ± 7.7** | 22.8 ± 2.7 | 83.4 ± 26.1*** |
| Pelvic thrusts | 531.2 ± 83.2 | 1026.2 ± 150.5* | 560.3 ± 100.6 | 995.1 ± 129.8* |

Quantification of sexual behavior components displayed by intact sexually naive and experienced control (ERα$^{fl/fl}$) and mutant (ERα$^{NesCre}$) males. Data are means ± S.E.M. for 10 males per genotype.
*$p < 0.05$, **$p < 0.01$; ***$p < 0.001$ versus control littermates.

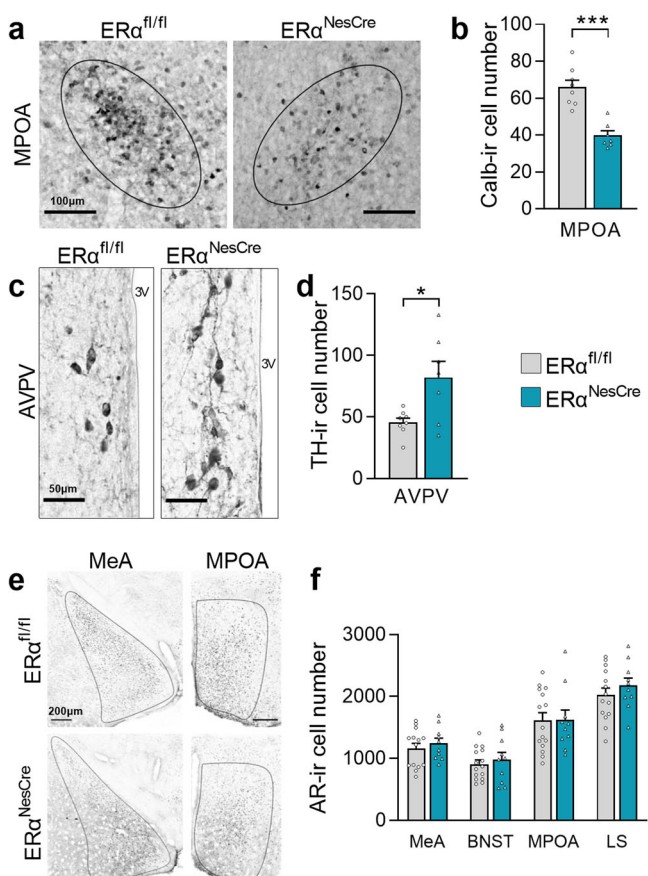

**Fig. 4 Neural *ERα* deletion altered the organization of sexually dimorphic populations.** **a** Representative calbindin-immunoreactivity (ir) in the medial preoptic area (MPOA) of control (ERα$^{fl/fl}$; *left*) and mutant males (ERα$^{NesCre}$; *right*). Scale bar 100 μm. **b** Quantification of calbindin-ir neurons in control and mutant males ($n = 7$–8 per genotype), ***$p < 0.001$ vs controls. **c** Representative images of tyrosine hydroxylase (TH)-ir in the in the anteroventral periventricular area (AVPV) of control (left) and mutant (right) males. Scale bar 50 μm. **d** Quantification of TH-positive neurons in the AVPV, *$p < 0.05$ versus controls ($n = 7$–8 per genotype). **e** Representative androgen receptor (AR)-ir in the medial amygdala (MeA; left) and the MPOA (right). Scale bars = 200 μm. **f** Quantification of the number of AR-ir neurons in the MeA, bed nucleus of stria terminalis (BNST), MPOA, and lateral septum (LS) in controls and mutants ($n = 10$–15 males per genotype); $p = 0.3$ and 0.9, respectively. All data are shown as means ± S.E.M. with individual values.

| Table 2 Sexual behavior of castrated and testosterone supplemented ERα$^{NesCre}$ males. | | |
|---|---|---|
| | **ERα$^{fl/fl}$** | **ERα$^{NesCre}$** |
| Seminal vesicle weight (%bw) | 1.44 ± 0.12 | 1.58 ± 0.11 |
| Latency to behavior (min) | | |
| Mounts (M) | 2.32 ± 0.26 | 16.73 ± 6.70* |
| Mounts with intromissions (MI) | 2.85 ± 0.27 | 17.40 ± 7.22* |
| Ejaculation | 46.26 ± 12.34 | 160.23 ± 44* |
| Mating duration (min) | 43.8 ± 12.35 | 143.50 ± 47.05* |
| Number of events | | |
| Mounts (M) | 8.63 ± 2.79 | 6.86 ± 2.72 |
| Mounts with intromissions (MI) | 15.75 ± 2.91 | 12.71 ± 4.38 |
| Pelvic thrusts | 672.75 ± 107.72 | 603.43 ± 148.24 |
| Intromission ratio MI/M + MI (%) | 68.01 ± 8.18 | 67.26 ± 3.68 |

The mean weight ± S.E.M of seminal vesicles is indicated as percentage of body weight (bw) for control (ERα$^{fl/fl}$) and mutant (ERα$^{NesCre}$) males, which were gonadectomized and normalized for their testosterone levels (9–10 males per genotype). The latencies and behavioral events are indicated for sexually experienced males that reached ejaculation (7–8 males per genotype). Data are means ± S.E.M. *$p < 0.05$ versus control littermates.

males compared to their control littermates. Immunohistochemical analysis of the AR showed a comparable number of AR-immunoreactive neurons in control and mutant males in the medial amygdala, bed nucleus of stria terminalis, medial preoptic area, and lateral septum (Fig. 4e, f).

The unchanged number of AR-immunoreactive neurons in key regions underlying sexual behavior, together with the increased levels of circulating testosterone may compensate for the sexual dysfunction caused by neural *ERα* deletion. We thus analyzed sexual behavior of control and mutant males, which were gonadectomized and supplemented with equivalent levels of testosterone. Table 2 shows that under these normalized hormonal conditions, evidenced by the comparable relative weight of seminal vesicles, the latencies to initiate the first mount and intromission and to reach ejaculation were higher in sexually experienced mutants (7.2-, 6-, and 3.5-fold above controls, respectively). In contrast, no differences were found between the two genotypes in the number of mounts, intromissions, and pelvic thrusts.

In continuous fertility tests with control females, neural *ERα* knockout males were able to produce offspring. The latency to produce the first litter (20.33 ± 0.33 days versus 27.67 ± 0.67 days for controls), the number of litters (7.33 ± 0.29 versus 7.33 ± 0.33 for controls) and litter size (6.26 ± 0.93 versus 6.77 ± 0.76 for controls) were comparable between the two genotypes.

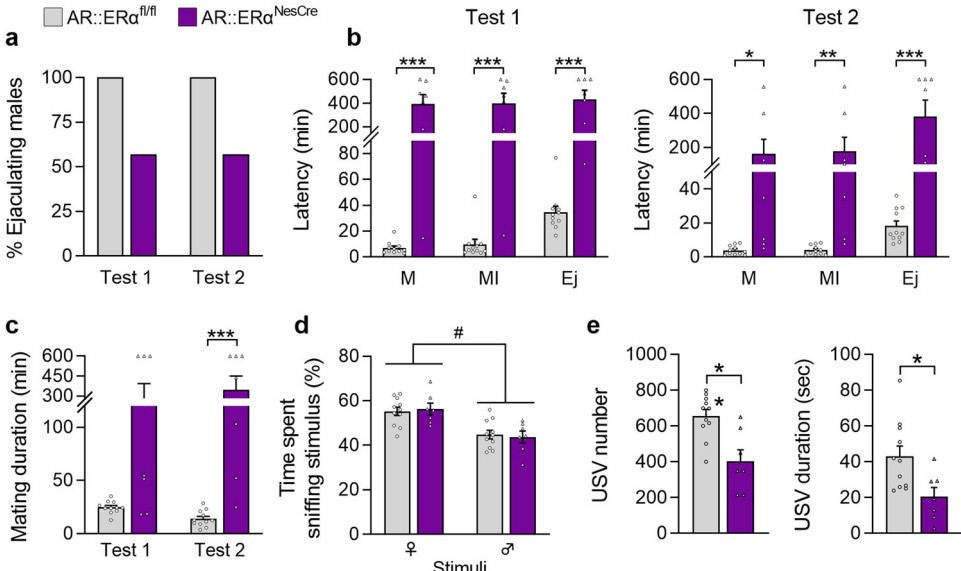

**Fig. 5 The behavioral effects of double *AR::ERα* deletion in males. a** Percentage of control (AR::ERα^fl/fl) and mutant (AR::ERα^NesCre) males reaching ejaculation in tests 1 (naive animals) and 2 (sexually experienced animals). Data are shown as percentages. **b** Behavioral components of sexual behavior for control and mutant males (n = 7–11 per genotype). Latencies to mount (M), intromit (MI) and to reach ejaculation (Ej) in Tests 1 and 2; *p < 0.05, **p < 0.01, ***p < 0.001 versus controls. **c** Mating duration from the first mount until ejaculation; p = 0.2 for Test1 and ***p < 0.001 for Test 2 (n = 7–11 per genotype). **d** Olfactory preference of sexually experienced controls and mutants (n = 7–11 per genotype) expressed as the percentage of total time spent sniffing stimuli. The effect of stimulus (#p < 0.05) is indicated. **e** Total number (left) and duration (right) of ultrasonic vocalizations (USV) produced by controls and mutants (n = 7–11 per genotype) in the presence of a sexually receptive female during a 4-min test. **p < 0.01 and *p < 0.05 versus control littermates. Instead stated otherwise, data are shown as means ± S.E.M. with individual values.

**Table 3 Sexual behavior of AR::ERα^NesCre males.**

| Behavioral events | Sexually naive (Test 1) | | Sexually experienced (Test 2) | |
|---|---|---|---|---|
| | AR::ERα ^fl/fl | AR::ERα ^NesCre | AR::ERα ^fl/fl | AR::ERα ^NesCre |
| Number of Mounts (M) | 8.27 ± 1.90 | 8.57 ± 2.57 | 6.72 ± 1.57 | 30.85 ± 9.18** |
| Number of Mounts with intromissions (MI) | 21.0 ± 1.99 | 10.28 ± 3.44* | 14.18 ± 2.33 | 23.71 ± 5.42 |
| Intromission ratio MI/M + MI (%) | 72.51 ± 4.23 | 44.9 ± 9.04** | 69.19 ± 4.69 | 46.28 ± 7.21* |
| MI duration (s) | 24.71 ± 2.35 | 19.19 ± 4.70 | 28.51 ± 3.61 | 22.03 ± 5.77 |
| MI interval (s) | 37.82 ± 5.75 | 6125.97 ± 5012.20 | 24.20 ± 3.24 | 387.52 ± 131.23** |
| Pelvic thrusts | 570.63 ± 49.05 | 153.57 ± 49.65*** | 443.32 ± 46.65 | 357.14 ± 98.74 |

Quantification of sexual behavior components displayed by sexually naive and experienced control AR::ERα^fl/fl and mutant AR::ERα^NesCre males. Data are means ± S.E.M. for 7–11 males per genotype. *p < 0.05, **p < 0.01, ***p < 0.001 versus control littermates.

**Double neural *AR* and *ERα* knockout severely impaired male sexual behavior in mice.** In previous studies, we have shown that AR^NesCre deletion greatly impaired but did not abolish the expression of male sexual behavior[8,11]. We thus determined the impact of neural deletion of both *AR* and *ERα* on male behaviors. In this double knockout model, only 57% of AR::ERα^NesCre mutant males versus 100% of controls reached ejaculation in tests 1 and 2 (Fig. 5a). In addition, the latencies to initiate the first mount and intromission and to reach ejaculation were greatly increased in test 1 for mutants (65-fold, 58-fold, and 14-fold above controls, respectively) (Fig. 5b). There was no significant amelioration in Test 2 (48-fold, 51-fold, and 24-fold above controls, respectively). In accordance, mating duration was significantly increased in both Tests 1 and 2 (12-fold and 28-fold above controls, respectively) (Fig. 5c). Quantification of the behavioral events showed reduced number of intromissions and thrusts in test 1, a higher number of mounts in test 2 and a reduced intromission ratio and a higher intromission interval for mutants in both tests (Table 3).

This sexual alteration was associated with a normal olfactory preference towards female stimuli (Fig. 5d) but a reduced total number and duration of emitted syllables (−43% and −51% versus controls, respectively) (Fig. 5e).

Double neural AR::ERα^NesCre males had a lower body weight (−15%) and a higher relative weight of seminal vesicles (+53%) and circulating levels of testosterone (+100%) in comparison with their control littermates (Supplementary Table 2).

**Effects of neural *ERα* or *AR::ERα* deletion on other behaviors.** Male and female mice, subjected to sexual behavior tests, were also analyzed for locomotor activity and anxiety-state level. These latter behaviors are also regulated by estrogens and could interfere with the expression of sexual behavior if altered. Ovariectomized and hormonally primed females carrying the ERα^NesCre mutation and gonadally intact ERα^NesCre males displayed a comparable activity to their respective control littermates in the circular corridor over the 120-min test (Fig. 6a, c). Similarly, no differences were found between control and mutant

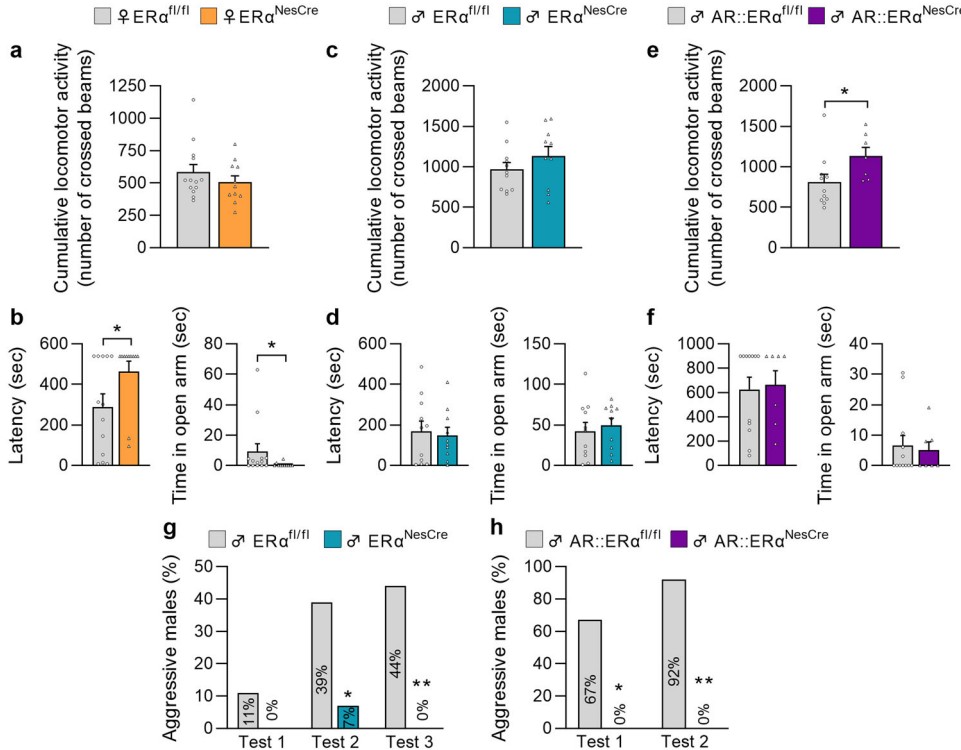

**Fig. 6 Effect of neural ERα or AR::ERα deletion on locomotor activity, anxiety-related, and aggressive behavior. a, c, e** Cumulative locomotor activity recorded in a circular corridor during the 2 h-test for control (ERα$^{fl/fl}$) and mutant (ERα$^{NesCre}$) females (**a**, $n = 11$–13 per genotype), control (ERα$^{fl/fl}$) and mutant (ERα$^{NesCre}$) males (**c**, $n = 10$–11 per genotype), or control (AR::ERα$^{fl/fl}$) and mutant (AR::ERα$^{NesCre}$) males (**e**, $n = 7$–11 per genotype). **b, d, f** Anxiety-related behavior measured in the elevated O-maze for control (ERα$^{fl/fl}$) and mutant (ERα$^{NesCre}$) females (**b**, $n = 11$–13 per genotype), control (ERα$^{fl/fl}$), and mutant (ERα$^{NesCre}$) males (**d**, $n = 10$-11 per genotype), or control (AR::ERα$^{fl/fl}$) and mutant (AR::ERα$^{NesCre}$) males (**f**, $n = 7$-11 per genotype). The latency to enter the open arm (left) and the time spent in the open arm (right) were analyzed. **g, h** Percentage of males from the ER$^{NesCre}$ (**g**) or AR::ERα$^{NesCre}$ (**h**) mouse lines displaying aggressive behavior towards non-aggressive males in the resident-intruder test across two or three tests. *$p < 0.05$, **$p < 0.01$ versus the control group. Data are shown as means ± S.E.M. with individual values (panels **a–f**) or as percentage (panels **g, h**).

ERα$^{NesCre}$ males, which were castrated and supplemented with equivalent testosterone concentrations (Supplementary Figure 2a). In contrast, a higher locomotor activity (+44%) was observed in intact males carrying the double AR::ERα$^{NesCre}$ mutation compared to their control littermates (Fig. 6e).

The anxiety-state level assessed in the O-Maze was increased in ERα$^{NesCre}$ females as evidenced by the higher latency to enter the open arms and reduced time spent in the open arms compared to their control littermates (Fig. 6b). No changes in the anxiety state level were observed in intact ERα$^{NesCre}$ males (Fig. 6d), castrated and testosterone-supplemented ERα$^{NesCre}$ mice (Supplementary Fig. 2b), or AR::ERα$^{NesCre}$ animals (Fig. 6f) compared to their corresponding control littermates.

Assessment of aggressive behavior showed that it was completely impaired in both neural ERα and AR::ERα knockout males as shown by the very low percentage or absence of mutants showing this behavior towards non-aggressive intruders compared to their corresponding control littermates (Fig. 6g, h).

## Discussion

The present study shows that neural ERα deletion completely inhibited female sexual behavior and greatly impaired the gonadotropic axis, confirming the predominant neural role of ERα in female reproduction. In comparison, neural ERα deletion impaired the expression of male sexual behavior and organization of sexually dimorphic neuronal populations in the preoptic area. Nevertheless, mutant males were able to achieve mating and produce offspring. A greater alteration of sexual behavior was

observed when both neural ERα and AR were deleted, highlighting the complementary roles of both receptors in male behavior.

Neural ERα knockout female mice had an arrested estrous cycle, with uterine hypertrophy and increased estradiol levels and were infertile as evidenced by the absence of corpora lutea. In mutant females, an altered pubertal maturation with an advanced time of vaginal opening and first estrus was also observed (Supplementary Fig. 3). These data are in agreement with the phenotype reported for females lacking ERα in neurons[16,17], or in Kiss1-[22] and Tac2-expressing cells[21].

At the behavioral level, under normalized hormonal levels between the two genotypes, mutant females were unable to adopt a lordosis posture, despite normal olfactory preference towards males. The comparable loss of lordosis behavior previously reported for global ERα knockout females[30] can thus be explained at least in part by the altered ERα signaling pathway at the neural level.

The deficient sexual behavior was probably due to the much lower number of PR-expressing neurons in the ventromedial hypothalamus, possibly due to the lack of estrogen-induced up-regulation of PR expression in the absence of neural ERα. Indeed, female receptivity during the estrus stage is triggered by estrogen-induced up-regulation of PR in rodents (for review: Mhaouty-Kodja et al.[31]).

The NesCre transgene used in the present study drives gene excision as early as embryonic day 10.5. It is thus possible that this early ERα deletion resulted in a deficient perinatal organization of brain areas controlling both the expression of sexual

behavior and also the gonadotropic axis, thus leading to this infertile phenotype. Indeed, changes in the number of kisspeptin and TH neurons of the medial preoptic area tend to favor a lack of feminization of this structure in mutant females. In comparison, neither a mutation preventing membrane ERα signaling[32] nor neural *ERα* deletion[14] were found to have any effect on the expression of female sexual behavior or the organization of related brain structures. Together, these observations support the idea of the predominant role played by nuclear ERα and its related genomic pathways in the neural control of female sexual behavior and gonadal axis function.

In male mice, neural *ERα* deletion resulted in increased latencies to initiate and achieve sexual behavior. In particular, sexual experience ameliorated the latency to mount and intromit but not the time needed to reach ejaculation. This moderate deficiency along with normal olfactory preference and slightly impaired emission of ultrasonic vocalizations could explain the fertile phenotype of mutant males. Indeed, neural *ERα* mutants were able to produce offspring like their control littermates. These results contrast with the previously reported phenotype of global *ERα* knockout males, which were infertile and exhibited greatly reduced sexual behavior[33–35]. It is possible that these huge changes in global knockout males were due to cumulative effects of *ERα* deletion in neural and peripheral sites; ERα plays a key function in the male urogenital tract.

Sex steroid hormones permanently organize the sexually dimorphic neuronal populations of the preoptic nucleus during the perinatal period. The lower number of calbindin-immunoreactive neurons and higher number of TH-immunoreactive neurons in neural *ERα* knockout males are in favor of an altered perinatal masculinization/defeminization processes in mutant males. It is possible that the consequences of altered brain organization and resulting sexual dysfunctions were partly compensated by the increased levels of testosterone, due to the altered negative feedback exerted by testosterone and its neural estradiol metabolite and unchanged number of AR-immunoreactive neurons in the neural circuitry controlling sexual behavior. When testosterone levels were normalized between the two genotypes, ERα[NesCre] males still exhibited an altered sexual behavior but the behavioral differences as compared with controls were reduced.

We have previously reported a greater impairment of sexual behavior in males lacking the neural *AR* as evidenced by the observed higher latencies to initiate and achieve sexual behavior and a more important reduction in the emission of ultrasonic vocalizations together with an hypofertile phenotype[8,10,11,36]. In these mutant males, no effect of neural *AR* deletion was observed on sexually dimorphic populations of the preoptic area[10,37] but changes in cell number, soma size, or dendritic arborization were observed in spinal nuclei involved in erection and ejaculation[11,12,37]. Together, these observations support the complementary roles of AR and ERα in the control of sexual behavior in male mice. Accordingly, double knockout males lacking both the *AR* and *ERα* in the nervous system displayed a more important sexual deficiency than neural *ERα* knockouts. AR::ERα[NesCre] males took much longer to initiate mounts and exhibited reduced emission of ultrasonic vocalizations, suggesting an impaired motivational state. In addition, only half of them reached ejaculation and those achieving mating took a very long time in comparison with control littermates with no behavioral improvement after a first sexual experience although olfactory preference remained unchanged. Altogether, these data strongly suggest that they possibly also displayed an impaired erectile activity and organization and activation of the spinal nucleus of the bulbocavernosus as we have previously shown for male mice lacking the neural *AR*[8,11]. Further studies will address the

reproductive phenotype of AR::ERα[NesCre] males in more detail to determine whether the ejaculating males are able to produce litters in continuous fertility tests.

In chemosensory areas located downstream of the olfactory bulb, study is still required of whether AR and ERα operate in identical or different neuronal populations. In this context, a previous study showed that *ERα* deletion in inhibitory Vgat neurons impaired mating[23]. In addition, a mutation preventing membrane ERα signaling triggered altered behavior and reduced number of calbindin- and kisspeptin-ir neurons[32], thus suggesting the involvement of different ERα signaling pathways in these processes.

Neural *ERα* deletion had no effect on locomotor activity in females. In contrast, it increased their anxiety-related behavior compared to their control littermates. This may participate to their inhibited sexual behavior. Comparison with the previous effects for neural *ERβ* on mood behavior[13,14] suggests that estradiol acts through both ERα and ERβ signaling pathways to modulate this behavior.

In males, assessment of the effects of neural *ERα* or *AR::ERα* deletion on locomotor activity showed genotype effects only on the activity of AR::ERα[NesCre] males, which was elevated maybe due to the higher sex steroid levels of these mutants compared to their control littermates. Indeed, mutant AR[NesCre] male mice displayed also a high activity that was restored to control levels by gonadectomy and supplementation with equivalent concentration of testosterone[8]. No effects of neural *ERα* or *AR::ERα* deletion were observed on anxiety-related behavior. Estrogens may thus regulate differently the basal anxiety state level between males and females. In males, if estrogens seem to be less involved in the modulation of mood behavior, they regulate together with androgens the expression of social behavior such as aggression. Indeed, suppression of one of the three neural estrogen or androgen pathways (ERα, ERβ, or AR) has been found to be sufficient to greatly impair or completely inhibit the expression of aggressive behavior (refs. [7,8,13], present study).

In summary, the present data show that neural *ERα* deletion triggers sex differential effects on reproductive behavior. In females, neural ERα is mandatory for the organization and activation of structures underlying sexual behavior. In males, although ERα plays a role in the early neuroanatomical organization of brain areas involved in sexual behavior, both AR and ERα are required for an efficient behavior. These observations were confirmed by the sexual phenotype of double knockout males lacking both receptors in the nervous system.

Neural *ERα* deletion was also found to affect anxiety-related behavior in females but not in males under non-stressful basal conditions. The data suggest a participation of neural ERα signaling pathway, as for neural ERβ[13,14], in estrogen-dependent modulation of anxiety-related behavior. In males, while the neural contribution of ERα pathway seems less important in the regulation of anxiety-related behavior, both pathways together with ERβ[13] are important for the expression of aggressive behavior. It would be interesting for further studies to identify the neuronal versus glial contribution of these signaling pathways across life stages.

## Methods

**Animals**. The ERα[NesCre] mouse line was obtained, on a mixed C57BL/6 x CD1 background, by crossing floxed *ERα* females in which the exon 3 of *ERα* was flanked by loxP sites[38] with floxed *ERα* males expressing the Cre recombinase under the control of the rat Nestin promoter (NesCre transgene; [39]). Mutant mice (ERα[fl/fl], NesCre: ERα[NesCre]) and their control littermates (ERα[fl/fl]) were used. To obtain the double AR::ERα[NesCre] mouse line, we first generated animals carrying floxed *ERα* and *AR* genes by crossing ERα[fl/fl] and AR[fl/fl] mouse lines. Homozygous floxed females (*ERα*[fl/fl]::*AR*[fl/fl]) were then crossed with ERα[NesCre] males to generate mutant males (ERα[fl/fl]::AR[fl/Y], NesCre: AR::ERα[NesCre]) and their control littermates

(ERα^fl/fl::AR^fl/Y; AR::ERα^fl/fl). After weaning, mice were kept at 22 °C under an inverted light schedule, i.e. the dark time began at 1:30 PM with a 12:12 h light-dark cycle and fed a standard diet with free access to food and water.

All experiments were conducted on adult animals of 3 to 6 months of age, in compliance with the French and European legal requirements (Decree 2010/63/UE) and were approved by the "Charles Darwin" Ethical committee (project number 01490-01).

**PCR and quantitative RT-PCR.** The selective neural *ERα* deletion was assessed by PCR and qRT-PCR as previously described[8,38]. For PCR, control and mutant females were sacrificed and brain and ovaries were collected. Detection of the Cre transgene and the floxed or truncated *ERα* allele was carried out using the following primers: ERα forward: 5′-AGG-CTT-TGT-CTC-GCT-TTC-C-3′; ERα reverse: 5′-GAT-CAT-TCA-GAG-AGA-CAA-GAG-GAA-CC-3′; NesCre forward: 5′-GCC-T GC-ATT-ACC-GGT-CGA-TGC-AAC-GA-3′; reverse: 5′- GTG-GCA-GAT-GGC-GCG-GCA-ACA-CCA-TT-3′. Amplification conditions were: 5 min at 95 °C for denaturation, 40 cycles comprising 30 s at 95 °C for denaturation, 30 s at 58 °C for annealing and 1 min at 72 °C for elongation and then 10 min at 72 °C for final elongation.

For semi-quantitative RT-PCR, ovaries, testes and three brains areas (cortex, hippocampus, hypothalamus) were collected and rapidly frozen with dry ice. Total RNAs were extracted using TRIzol reagent (Invitrogen, Carlsbad, USA) and 1 µg was reverse transcribed into cDNA using the Superscript III First-Strand Synthesis System (Invitrogen Carlsbad, USA) and random hexamers. PCR reactions were performed using the resulting cDNA (4 µl), dNTPs (10 nM each), primers (100 µM, Eurogentec, Seraing, Belgium) and Taq DNA pol (Invitrogen) in a MyCycler Thermal Cycler (Bio Rad, Marne la Coquette, France). Primers for ERα (forward: 5′-CGT-GTG-CAA-TGA-CTA-TGC-CTC-T-3′; reverse: 5′- TGG-TGC-ATT-GGT-TTG-TAG-CTG-G -3′) and GADPH (forward: 5′- TGC-ACC-ACC-AAC-TGC-TTA-GC -3′; reverse: 5′- GGC-ATG-GAC-TGT-GGT-CAT-GAG -3′) were used. Amplification conditions were: 5 min at 95 °C for denaturation, 40 cycles comprising 30 s at 95 °C for denaturation, 30 s at 56 °C for annealing and 1 min at 72 °C for elongation and then 10 min at 72 °C for final elongation. The amplified cDNA fragments were separated by electrophoresis in a 1.5% agarose gel and stained by ClearSight DNA stain (Euromedex-BM, Souffelweyersheim).

**Reproductive physiology.** Prepubertal females were monitored daily for visual evidence of vaginal opening from postnatal day 15 and vaginal smears flushed with physiological saline were taken daily and colored with hematoxylin-eosin to identify the estrus stage based on cell types[14,40]. Adult females were also examined daily for the estrous cycle for 3 weeks using the same protocol, and were sacrificed at the diestrus phase to collect trunk blood and weigh ovaries and uteri. Ovaries were fixed in Bouin's buffer, washed in ethanol, paraffin-embedded, and sliced into 5 µm sections[40]. The sections were mounted and deparaffinized, stained with hematoxylin–eosin and the number of corpus lutea was analyzed.

At sacrifice of males, blood was collected and body, testis, and seminal vesicles were weighed. The epididymis was dissected and sperm collected, incubated in PBS for 10 min at 37 °C, and centrifuged 5 min at 100×g. The supernatant was removed and 10-fold diluted[8] and spermatozoa were counted in a hemocytometer with a Normaski microscope. Fertility was assessed in continuous mating for 5 months. Each male was paired with two age-matched control females. The interval from mating to the first litter and the number and size of litters were reported.

Sera obtained by decantation and centrifugation (2500 × g for 15 min) were used to measure by radioimmunoassay circulating levels of estradiol using DSL-4800 kit (Beckman-Coulter), and testosterone by the platform of the laboratory of Behavioral and Reproductive Physiology (UMR 0085 INRAE-CNRS- IFCE-University of Tours).

**Experimental conditions for behavioral tests.** Assays were conducted under red-light illumination 2 h after lights-off and were videotaped for later analyses. The devices used in the tests, with the exception of animal home cages, were cleaned with 10% ethanol between trials. The analyses were performed by blind observation.

Stimulus C57BL/6J females used for the analyses of male behaviors and control (ERα^fl/fl) and mutant (ERα^NesCre) females were ovariectomized under general anesthesia (xylazine 10 mg/kg, ketamine 100 mg/kg), and implanted with 1 cm sc SILASTIC implants (3.18 mm outer diameter X 1.98 mm inner diameter; Dow Corning, Midland, MI) filled with 50 µg of estradiol-benzoate (Sigma-Aldrich) diluted in 30 µl of sesame oil. Females were subcutaneously injected with 1 mg of progesterone (Sigma-Aldrich) in 100 µl of sesame oil 4 to 5 h before the tests to experimentally induce their receptivity. The stimulus C57BL/6J females used for the analyses of male behaviors were all sexually experienced twice and screened before each mating test for their lordosis behavior with stud males.

Control (ERα^fl/fl) and mutant (ERα^NesCre) males were gonadectomized under similar experimental conditions and implanted with a subcutaneous SILASTIC tubes containing 10 mg of testosterone (Sigma–Aldrich). Stimulus C57BL/6J males used for the analyses of female behaviors were sexually experienced and selected for showing a strong mounting behavior in less than 5 min after the presentation of a female (Supplementary Fig. 4).

**Sexual behavior.** Sexually naive females were tested for their sexual receptivity toward male mounting as previously described[14]. Test began when the female was introduced into the home cage of a sexually experienced C57BL/6J stud male and lasted for 20 min or until the female received 20 mounts. Females were tested three times with an interval of 1 week between tests. The lordosis quotient (number of lordosis responses/number of mounts) was reported.

Sexually naive males were housed in individual cages 4 days before the beginning of the tests. Each male was tested in its home cage for 10 h after the introduction of a sexually experienced stimulus female. Males were tested twice in tests 1 (naive) and 2 (sexually experienced) with a time interval of one week. Male sexual behavior was analyzed by scoring the latency to initiate mounts, intromissions and to reach ejaculation, and the frequency of mounts, intromissions, thrusts as previously described[8]. Mating duration was defined as the time from the first mount to ejaculation and the ratio intromission as the proportion of mounts with successful intromission. A latency of 600 min was given to males that did not initiate the behavior.

**Olfactory preference.** Sexually experienced males and females were tested for odor preference in a Y-maze apparatus as previously described[10,14,41]. Female mice were allowed to become familiar with the maze for 10 min over two consecutive days. On the experimental test day, each Y-arm randomly received either an anesthetized stimulus male or female. Subjects were free to investigate the stimuli for 9 min. Time spent in investigation by the tested animals was recorded and expressed as a percentage of the total time of investigation.

**Ultrasonic vocalizations.** Each sexually experienced male was tested in its home cage in the presence of a receptive female. The vocalizations were recorded for 4 min, after the introduction of a female, with an Ultrasound gate microphone CM16/CMPA (Avisoft Bioacoustics), connected to an Ultrasound recording interface plugged into a computer equipped with the Avisoft-SASLab Pro 5.2.09 recording software. The recordings were analyzed using SASLab Pro (Avisoft Bioacoustics). A cutoff frequency of 30 kHz was used and elements were separated based on an automatic threshold with a hold time of 15 ms[36,42]. The total number and duration of total ultrasonic vocalizations and syllables were reported.

**Locomotor activity.** Activity was analyzed in a computed circular corridor made up of two concentric cylinders and crossed by four diametrically opposite infrared beams (Imetronics) as previously described[8,14]. The locomotor activity was counted when animals interrupted two successive beams and had thus traveled a quarter of the circular corridor. The cumulative activity was reported.

**Anxiety-related behavior.** The anxiety state was assessed in the elevated O-maze paradigm as previously described[14]. The test lasted 9 min and began when the animal was placed in a closed arm. The latency to enter into an open arm (with the four paws out) and the time spent in the open arms were analyzed.

**Aggressive behavior.** Each sexually experienced male was tested in the resident-intruder test, after one week without bedding change, as previously described[7,8]. Tests lasted 10 min and began when a sexually inexperienced gonadally intact adult A/J male (The Jackson Laboratory, USA) was introduced into the home-cage of the tested resident male. Each resident male was exposed to a different intruder on three consecutive days. Aggressive behavior was defined as lunging, biting, wrestling and kicking. The proportion of males showing aggressive behavior was reported.

**Tissue preparation.** Brains from perfused animals were removed, postfixed overnight in 4% PFA, cryoprotected in 0.1 M phosphate buffer containing 30% sucrose and stored at −80 °C. Coronal sections (30 µm) were then processed for immunolabeling as previously described[10,14].

**AR, TH, and Calbindin immunostainings.** Endogenous peroxidases were blocked for 30 min in PBS containing 3% $H_2O_2$ and sections were incubated in blocking solution (PBS containing 0.2% Triton X-100 and 1% bovine serum albumin for AR or 5% horse serum for TH and Calbindin) 2 h at room temperature (RT). Sections were incubated at 4 °C in the blocking solution supplemented with 1:200 diluted rabbit anti-AR (sc-816, Santa Cruz Biotechnology) for 36 h, or with 1:2000 diluted mouse anti-TH antibody (MAB318, Millipore) or 1:1000 diluted mouse anti-Calbindin antibody (C9848, Sigma-Aldrich) overnight. Sections were incubated in PBS-0.2%Triton solution supplemented with 1:1000 diluted biotinylated secondary antibody (goat anti-rabbit BA1000, Vector; or horse anti-mouse BA2000, Vector) for 2 h at RT. Signals were amplified with VECTASTAIN® ABC kit (Vector) 1:1000 diluted in PBS for 1 h at RT and visualized with 3,3′-diaminobenzidine tetra-hydrochloride (DAB 0.02%, 0.01% $H_2O_2$ in 0.05 M Tris, pH 7.4). Sections were mounted on slides, dehydrated and coverslided using Permount (Sigma-Aldrich). The slices were scanned using an automatic slice scanner (NanoZoomer, Hamamatsu). Regions of interest were drawn based on the Mouse Brain Atlas[43] and immunoreactive cells were manually counted using the NDPview software. AR positive nuclei quantification was performed bilaterally on one representative section of each area (MeA: plate 43; BNST-MPOA: plate 30; LS: plate 27). TH

positive cells quantification was performed bilaterally on five sections covering the entire AVPV/PeN (plates 28 to 33). Quantification of calbindin-positive cells was performed on one representative section of the SDN-POA (plate 33).

**Kisspeptin and PR labelings**. Sections were incubated in a blocking solution (PBS containing 0.3% Triton X-100 and 3% donkey serum) 2 h at RT and then at 4 °C for 36 h in the blocking solution supplemented with 1:10,000 diluted sheep anti-kisspeptin (AC053, generous gift of I. Franceschini[44]) or 1:100 rabbit anti-PR (A0098, Dako) antibodies. The sections were incubated in PBS-0.3%Triton solution supplemented with 1:500 donkey anti-sheep-Alexa 488 (A11015, Life Technologies) or 1:1000 biotinylated horse anti-rabbit (BA1100, Vector) secondary antibodies for 2 h at RT. The signal was amplified with a Cy3-Streptavidin (s32355, Life Technologies) 1:1000 diluted in PBS for 1 h at RT. The sections were counterstained in a 2 µg/ml Hoechst solution (Invitrogen) for 5 min, mounted on slides and coverslipped with Mowiol. Three sections covering the AVPV/PeN (plates 28-31 of the Paxinos and Franklin Mouse Brain Atlas) were imaged using an inverting microscope equipped with an apotome (Axiovert 200 M coupled to the AxioCam MRm camera, Zeiss). Z-stacks (3 µm) were carried out on the entire thickness of the sections. Kisspeptin positive cells were manually counted on using the ImageJ software (NIH). PR positive nuclei quantification was performed bilaterally on one representative section of the VMHvl (plates 45 of the Paxinos and Franklin Mouse Brain Atlas).

**Statistics and reproducibility**. Analyses were performed using GraphPad Prism 8.0 (GraphPad Software, Inc.) Assumption of normality (Shapiro-Wilk test) was tested before conducting the following statistical analyses. Two-way repeated measures ANOVA was used to analyze the effect of genotype and stimulus or genotype and time on olfactory preference, lordosis quotient for ERα^NesCre males and females and AR::ERα^NesCre males. One-way ANOVA with Tukey's post hoc was used to analyze group effect on kisspeptin-immunoreactivity for ERα^NesCre females. Mann–Whitney test was used to compare estrous cycle duration, the latency, number of entries and time spent in the O-maze open arm for ERα^NesCre females. It was also used to compare in males the latencies to mount and intromit, the number of mounts and pelvic thrusts in sexually naive and experienced ERα^NesCre males; the number of pelvic thrusts in sexually naive ERα^NesCre males and the interval between intromissions in sexually experienced ERα^NesCre males, the number of short, downward, modulated, and one-jump USV syllables, the duration of short, upward, downward, modulated, mixed and one-jump USV syllables, the levels of plasma testosterone and estradiol, the number of spermatozoa, the body weight and relative testis weight in ERα^NesCre males, as well as the latencies to mount, intromit, and ejaculate, the mating duration and the interval between intromissions in sexually naive and experienced AR::ERα^NesCre males, the latency to enter, the number of entries and the time spent in the O-maze open arm, the body weight and the relative testis weight in AR::ERα^NesCre males. The remaining data were analyzed by a two-tailed unpaired Student t-test. $P$ values less than 0.05 were considered statistically significant (*$p < 0.05$, **$p < 0.01$, and ***$p < 0.001$). Results are presented as means ± S.E.M., except for the percentage of ejaculating ERα^NesCre and AR::ERα^NesCre males and the percentage of aggressive ERα^NesCre and AR::ERα^NesCre males, which were analyzed by Fisher's exact test.

**Reporting summary**. Further information on research design is available in the Nature Research Reporting Summary linked to this article.

## Data availability
Source data is provided as Supplementary Data 1.

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

## Acknowledgements

We thank the Institut Biologie Paris-Seine platform for taking care of the animals. This work was funded by the Agence Nationale de la Recherche (ANR-09-CESA-006 and ANR-programme blanc SVSE 7-2012), CNRS, Inserm, and Sorbonne Université.

## Author contributions

S.M.K. and A.C.T. designed the study. S.R. provided the floxed ERα mouse line. A.C.T., S.D., L.N., D.C., and C.P. carried out the experiments. A.C.T., S.D., H.H.P., and S.M.K. analyzed the data. S.M.K. and A.C.T. wrote the manuscript and all authors read and approved the final manuscript.

## Competing interests

The authors declare no competing interests.
