## [Peer Review File · Communications Biology]

Reviewers' comments:

Reviewer #1 (Remarks to the Author):

Trouillet et al have provided a comprehensive characterization of physiological and behavioral phenotypes in female and male mice with a neural-specific deletion of ER α . In females, they report altered estrous cycle and a complete loss of sexual behavior. Mutant males show decreased mating behavior and aggression, and feminization of the MPOA and AVPV. The overall conclusions are consistent with central concepts in the field, namely that ER α is required to organize sex-specific circuitry and AR and ER α act together to facilitate the display of male-typical behaviors in adulthood. Although there are no surprises here, this work will be a useful reference for the field. The histological data add impact. The inclusion of additional behaviors beyond mating and aggression is interesting, however the data presentation of these assays needs improvement, see specific points below.

"Sexually dimorphic" should be removed from the manuscript title. It is a bit disingenuous to claim sexually dimorphic effects following loss of ER α when completely different parameters are assessed in the two sexes. Males don't have an estrous cycle and USVs were not measured in females so it's wrong to say that "ER α has sexually dimorphic effects" on these things. If something that appeared similar in controls was differentially affected in female and male mutants, then the authors could claim "sex differential effects of ER α deletion".

"The ovaries start liberating significant levels of estradiol around postnatal day 7." Please provide a citation for this statement. Citations 2-4 contain only behavioral data and do not report postnatal serum estradiol levels.

The introduction states that "studies conducted on males targeted life stages after the perinatal period". This is incorrect. Ref 22 assessed male behaviors using the same vGat-Cre and vGlut2-Cre drivers that were used in Ref 17. Both these drivers delete before birth, as is demonstrated by immunostaining for ER α at P0 in Figure 1 of Ref 22.

Figure 6: The control AR:ER α fl/fl mice show increased beam breaks, higher latency to enter open arms, decreased time in open arms, and increased aggression compared to control ER α fl/fl animals.

Similarly, male and female control ER α fl/fl animals show very different performance on the plus maze; males spend about 5x as much time in the open arm (compare 5B and 5D). Thus, it is not clear if "estrogens act in a sexually dimorphic manner to regulate the anxiety state level", as the authors state, or if these results merely illustrate underlying sex differences in assay performance. Groups should be included in the same statistical analyses if claims are made regarding their differences.

Reviewer #2 (Remarks to the Author):

This is an interesting paper focusing on topics of major biological and medical importance. It represents a lot of work and is well written. Genetic manipulations are well documented. The kisspeptin cell results are interesting, but I am a bit surprised by the results about TH cell numbers. Please explain better.

The lordosis behavior results replicate findings by Sonoko Ogawa., which perfectly match the progestin receptor results, as expected.

The authors' results in VMHvl and MPOA are fine, but they may also be interested in a small ER-expressing neuronal group medial to VMH but lateral to ARC.

The authors may be interested in a paper by Sonoko Ogawa in the journal Neuroendocrinology in the mid 1990s in which her genetic manipulations effectively reversed sex behavior phenotypes between males and females.

Are the results in Figure 5 due to motivational changes or sensorimotor incompetence ?

Reviewer #3 (Remarks to the Author):

The manuscript by Anne-Charlotte Trouillet et al. describes functions of neural ER α in the sexual differentiation of the brain and regulation of sex-specific social behaviors of rodents. This manuscript from Sakina Mhaouty-Kodja's lab is a continuation of a series of studies that analyzed the role of sex steroid receptors expressed in neurons in the control of reproductive behavior published by the same lab.

The authors generated neuron-specific ER α KO male and female mice and performed behavioral and histological analyses. The results revealed that neuronal ER α KO impairs estrous cycle, lordosis reflection, and feminization of the AVPV in female mice, and sexual behavior, aggressive behavior, and defeminization/masculinization of the MPOA and AVPV in male mice. Additionally, they mated the mice with a neuron-specific AR knockout mice, which had been previously reported by the authors, and found that males with both ER α and AR knockouts had larger impairments in sexual behavior. Therefore, these authors suggest roles of ER α in the regulation of reproductive function and sex-specific social behavior, and sex differentiation of the mouse brain.

This is an excellent manuscript on behavioral and histological analyses of the function of neuronal ER α in mice. I also think that the topic may be of interest to the field.

However, I have a few relatively minor comments, explained below.

Line 105, line 125, and line 162: In the results of the male and female sexual behavior tests, there is no description of the behavior of the stimulus animals. The conclusions would be strengthened by describing the behavior of both subjects and the stimulus animals to make it clear whether behaviors of the stimulus animals had any effect on the behavior of the subjects.

Line 203: The authors suggest that the morphological effects of ER α KO on the brain are due to the impaired formation of sex differences in the perinatal period. However, since the experiments in this manuscript only show knockout data, it is difficult to determine when ER α affects brain morphology and/or gene expression. Sex steroids have been reported to be involved in the formation of sex differences in the MPOA and AVPV not only in the perinatal period but also in the peripubertal period (Nat Neurosci, 2008,11(9):995-7. Endocrinology, 2017,158(10):3512-3525.). Additionally, the expression of Calbindin and Kiss1, which are the targets of immunostaining in this study, is regulated by sex steroids in the adult period (Brain Res Dev Brain Res. 2001, 129(2):125-33, Endocrinology, 2007, 148(4):1774-83.). Although I agree that aromatized testosterone in the perinatal period probably promotes brain masculinization via binding to neuronal ER α , the authors cannot exclude the possibility that neuronal ER α is necessary for brain sexual differentiation in other periods.

Line 358: The exact description of the SILASTIC implants is missing.

Line 430: The exact description of the brain map is missing.

Line 432: What is the meaning of "plate"?

Line 460: Both "naive" and "naïve" are used; the authors should maintain consistency in terminology usage.

Figure 2F: As far as I know, it is customary not to attach a unit to a lordosis quotient.

Reviewer #1:

Trouillet et al have provided a comprehensive characterization of physiological and behavioral phenotypes in female and male mice with a neural-specific deletion of ER α . In females, they report altered estrous cycle and a complete loss of sexual behavior. Mutant males show decreased mating behavior and aggression, and feminization of the MPOA and AVPV. The overall conclusions are consistent with central concepts in the field, namely that ER α is required to organize sex-specific circuitry and AR and ER α act together to facilitate the display of male-typical behaviors in adulthood. Although there are no surprises here, this work will be a useful reference for the field. The histological data add impact. The inclusion of additional behaviors beyond mating and aggression is interesting, however the data presentation of these assays needs improvement, see specific points below.

“Sexually dimorphic” should be removed from the manuscript title. It is a bit disingenuous to claim sexually dimorphic effects following loss of ER α when completely different parameters are assessed in the two sexes. Males don’t have an estrous cycle and USVs were not measured in females so it’s wrong to say that “ER α has sexually dimorphic effects” on these things. If something that appeared similar in controls was differentially affected in female and male mutants, then the authors could claim “sex differential effects of ER α deletion”.

We agree with your remark; the title was changed in line 1 to “Sex differential effects of neural estrogen receptor alpha deletion on reproductive behavior in mice” to indicate the different extent of effects of neural ER α deletion between males and females (inhibition of sexual behavior in females versus reduction in males).

This was also changed also in the conclusion (line 297).

“The ovaries start liberating significant levels of estradiol around postnatal day 7.” Please provide a citation for this statement. Citations 2-4 contain only behavioral data and do not report postnatal serum estradiol levels.

As you requested, a reference was added (line 43).

The introduction states that “studies conducted on males targeted life stages after the perinatal period”. This is incorrect. Ref 22 assessed male behaviors using the same vGat-Cre and vGlu2-Cre drivers that were used in Ref 17. Both these drivers delete before birth, as is demonstrated by immunostaining for ER α at P0 in Figure 1 of Ref 22.

We apologize for this mistake. The text was changed (lines 64-68): “In a male study, it has been shown that postnatal ER α deletion in vesicular GABA transporter (Vgat) neurons altered mating and territorial behaviors without changing hormonal levels [23], while its deletion in vesicular glutamate transporter (Vglut) 2 neurons increased testosterone levels but did not affect these behaviors. Other studies conducted on males targeted life stages after the perinatal period....”

Figure 6: The control AR:ER α fl/fl mice show increased beam breaks, higher latency to enter open arms, decreased time in open arms, and increased aggression compared to control ER α fl/fl animals. Similarly, male and female control ER α fl/fl animals show very different performance on the plus maze; males spend about 5x as much time in the open arm (compare 5B and 5D). Thus, it is not clear if “estrogens act in a sexually dimorphic manner to regulate the anxiety state level”, as the authors state, or if these results merely illustrate underlying sex differences in assay performance. Groups should be included in the same statistical analyses if claims are made regarding their differences.

Concerning the locomotor activity, we are grateful to the reviewer for pointing this discrepancy. The results presented for the AR::ER α ^{NesCre} mouse line corresponded to the total number of beam breaks,

whereas those for the $ER\alpha^{NesCre}$ mouse line corresponded to the number of times a mouse crossed two consecutive beams, reflecting the number of quarters traveled as explained in the Methods section. We apologize for this error, which was corrected in the revised Figure 6. This does not change the observed differences between $AR::ER\alpha^{fl/fl}$ controls and their mutant littermates (lines 190-191).

Otherwise, the females of the $ER\alpha^{NesCre}$ mouse line, the males of the $ER\alpha^{NesCre}$ line, and the males of the double knockout $AR::ER\alpha^{NesCre}$ line were tested separately and not at the same time for their locomotor activity or anxiety-related behavior. In addition, the $ER\alpha^{NesCre}$ females assessed in the behavioral tests were ovariectomized and hormonally primed as indicated in the text (185-188).

We therefore cannot include these groups in the same statistical analyses. For these reasons, the data were presented separately, and the effects of neural gene deletion were analyzed for each mouse line and sex by comparing mutants with their corresponding controls, since for each mouse line and sex control and mutant littermates were analyzed in the same experiments. What we then compared between the different mouse lines and sexes is the extent of effect of gene deletion, for example an increased anxiety state level in females versus no effects in males.

Changes were made in the revised manuscript in order to clearly specify that the comparisons were made between control and mutant littermates from the same mouse line and sex (lines 185-200).

Furthermore, the corresponding discussion was also re-written (lines 280-291).

Reviewer #2:

This is an interesting paper focusing on topics of major biological and medical importance. It represents a lot of work and is well written. Genetic manipulations are well documented.

The kisspeptin cell results are interesting, but I am a bit surprised by the results about TH cell numbers. Please explain better.

The number of TH-ir neurons was quantified in control females at metestrus-diestrus and proestrus-estrus stages, and in mutant females. The statistical analysis showed an effect with differences seen only between controls at metestrus-diestrus and mutant females. In fact, there were no significant differences between control females at metestrus-diestrus and proestrus-estrus stages as previously reported by others (Clarkson and Herbison, *J Neuroendocrinol* 2011; Leite et al., *J Neuroendocrinol* 2008).

Therefore, for a better clarity, we grouped the data for control females at metestrus-diestrus and proestrus-estrus stages since no significant differences were seen, to illustrate only the genotype effect (see the revised Figure 2D-E).

This paragraph was re-written in the revised manuscript (lines 110-114) and the figure legend modified (lines 646-654).

The lordosis behavior results replicate findings by Sonoko Ogawa., which perfectly match the progestin receptor results, as expected.

The authors' results in VMHv1 and MPOA are fine, but they may also be interested in a small ER-expressing neuronal group medial to VMH but lateral to ARC.

We guess that the reviewer refers to the sagittalis nucleus of the hypothalamus, which was identified in rats by Mori et al. (*PNAS* 2008). To our knowledge, it is not clear if such a nucleus is also present in mice. Furthermore, it is not possible to make a comparison between controls and mutants on the basis of ER α immunoreactivity in the neural ER α knockouts we used.

The authors may be interested in a paper by Sonoko Ogawa in the journal *Neuroendocrinology* in the mid 1990s in which her genetic manipulations effectively reversed sex behavior phenotypes between males and females.

A reference and comparison with the article of Ogawa et al. (1996) was added in the discussion section (lines 219-220: "The comparable loss of lordosis behavior previously reported for global ER α knockout females [30] can thus be explained at least in part by the altered ER α signaling pathway at the neural level."

Are the results in Figure 5 due to motivational changes or sensorimotor incompetence?

AR::ER α NesCre males took much longer to initiate mounts and exhibited reduced emission of ultrasonic vocalizations, suggesting an impaired motivational state. In addition, only half of them reached ejaculation and those achieving mating took a very long time in comparison with their control littermates with no behavioral improvement after a first sexual experience although olfactory preference remained unchanged. Altogether, these data strongly suggest that they probably also displayed an impaired erectile activity and organization and activation of the spinal nucleus of the bulbocavernosus as shown in our previous studies for male mice lacking the neural AR (Raskin et al., 2009; Raskin et al., 2012).

This was added in the revised manuscript (lines 264-270).

Reviewer #3:

The manuscript by Anne-Charlotte Trouillet et al. describes functions of neural ER α in the sexual differentiation of the brain and regulation of sex-specific social behaviors of rodents. This manuscript from Sakina Mhaouty-Kodja's lab is a continuation of a series of studies that analyzed the role of sex steroid receptors expressed in neurons in the control of reproductive behavior published by the same lab. The authors generated neuron-specific ER α KO male and female mice and performed behavioral and histological analyses. The results revealed that neuronal ER α KO impairs estrous cycle, lordosis reflection, and feminization of the AVPV in female mice, and sexual behavior, aggressive behavior, and defeminization/masculinization of the MPOA and AVPV in male mice. Additionally, they mated the mice with a neuron-specific AR knockout mice, which had been previously reported by the authors, and found that males with both ER α and AR knockouts had larger impairments in sexual behavior. Therefore, these authors suggest roles of ER α in the regulation of reproductive function and sex-specific social behavior, and sex differentiation of the mouse brain.

This is an excellent manuscript on behavioral and histological analyses of the function of neuronal ER α in mice. I also think that the topic may be of interest to the field.

However, I have a few relatively minor comments, explained below.

Line105, line 125, and line 162: In the results of the male and female sexual behavior tests, there is no description of the behavior of the stimulus animals. The conclusions would be strengthened by describing the behavior of both subjects and the stimulus animals to make it clear whether behaviors of the stimulus animals had any effect on the behavior of the subjects.

In our experimental conditions, the stimuli received a particular attention to ensure that the effects observed were mainly due to the altered behavior of the studied animals.

- Lordosis behavior of ER $\alpha^{fl/fl}$ and ER α^{NesCre} females was analyzed using sexually experienced C57BL/6J males, which were selected for showing a strong mounting behavior in less than 5 minutes after the presentation of a female. Furthermore, females of each genotype received a comparable number of mounts, as shown by the graph below.

- Male sexual behavior of ER $\alpha^{fl/fl}$ and ER α^{NesCre} males (or AR::ER $\alpha^{fl/fl}$ and AR::ER α^{NesCre} males) was analyzed using stimulus C57BL/6J females experimentally induced for their receptivity through ovariectomy and estradiol/progesterone priming. In addition, they were all sexually experienced twice with stud C57BL/6J males. Finally, before each behavioral test, females were all screened for their lordosis behavior with stud males. Only females showing a lordosis response after less than 3 mount attempts were used. Therefore, we never observed rejection behavior in our stimuli females.

Please see lines 370-377 of the revised manuscript, where we added this information.

Finally, the best controls of our experimental conditions are the floxed animals, which were used under similar experimental conditions. The comparison with their mutant littermates in several behavioral tests (mating of naive and experienced animals with a thorough analysis of the behavioral

components, olfactory preference, ultrasonic vocalizations) showed clear effects of genotypes, which were in accordance with the neuroanatomical analyses. Therefore, we do not think that adding more about the behavior of stimulus animals will add relevant points on the observed phenotypes.

Line 203: The authors suggest that the morphological effects of ER α KO on the brain are due to the impaired formation of sex differences in the perinatal period. However, since the experiments in this manuscript only show knockout data, it is difficult to determine when ER α affects brain morphology and/or gene expression. Sex steroids have been reported to be involved in the formation of sex differences in the MPOA and AVPV not only in the perinatal period but also in the peripubertal period (Nat Neurosci, 2008,11(9):995-7. Endocrinology, 2017,158(10):3512-3525.). Additionally, the expression of Calbindin and Kiss1, which are the targets of immunostaining in this study, is regulated by sex steroids in the adult period (Brain Res Dev Brain Res. 2001, 129(2):125-33, Endocrinology, 2007, 148(4):1774-83.). Although I agree that aromatized testosterone in the perinatal period probably promotes brain masculinization via binding to neuronal ER α , the authors cannot exclude the possibility that neuronal ER α is necessary for brain sexual differentiation in other periods.

This sentence “In comparison, neural ER α deletion impaired the expression of male sexual behavior and perinatal organization of sexually dimorphic neuronal populations in the preoptic area.” referred only to males with respect to the changed number of TH-ir and calbindin-ir cells in ER α ^{NesCre} males. However, we agree that the pubertal period is also an important organizational period. Therefore, in the revised manuscript, the reference to the period of brain masculinization was removed (line 206): “In comparison, neural ER α deletion impaired the expression of male sexual behavior and organization of sexually dimorphic neuronal populations in the preoptic area.”

Line 358: The exact description of the SILASTIC implants is missing.

The description of the implants was added in the revised manuscript (lines 366-367).

Line 430: The exact description of the brain map is missing.

The exact reference was added (line 444).

Line 432: What is the meaning of “plate”?

The mouse brain Atlas (Paxinos & Franklin, 2001) provides the coronal, sagittal and horizontal plates and corresponding diagrams or figures. Here, we refer to the exact term used in the Atlas for coronal sections.

Line 460: Both “naive” and “naïve” are used; the authors should maintain consistency in terminology usage.

Corrected throughout the text to “naive”.

Figure 2F: As far as I know, it is customary not to attach a unit to a lordosis quotient.

We changed the figure to express the lordosis quotient as a ratio without a unit. Consequently, values are comprised between 0 and 1 instead of 0 and 100. The main text was modified accordingly (line 118).

REVIEWERS' COMMENTS:

Reviewer #1 (Remarks to the Author):

I am satisfied with the response from the authors. With the rewrites they have made, I can now recommend this manuscript for publication.

Reviewer #3 (Remarks to the Author):

All the points raised in the first version of the manuscript have been properly addressed by the authors.

It is not necessary to give them further comments.